# Resectability and Ablatability Criteria for the Treatment of Liver Only Colorectal Metastases: Multidisciplinary Consensus Document from the COLLISION Trial Group

**DOI:** 10.3390/cancers12071779

**Published:** 2020-07-03

**Authors:** Sanne Nieuwenhuizen, Robbert S. Puijk, Bente van den Bemd, Luca Aldrighetti, Mark Arntz, Peter B. van den Boezem, Anna M. E. Bruynzeel, Mark C. Burgmans, Francesco de Cobelli, Marielle M. E. Coolsen, Cornelis H. C. Dejong, Sarah Derks, Arjen Diederik, Peter van Duijvendijk, Hasan H. Eker, Anton F. Engelsman, Joris I. Erdmann, Jurgen J. Fütterer, Bart Geboers, Gerie Groot, Cornelis J. A. Haasbeek, Jan-Jaap Janssen, Koert P. de Jong, G. Matthijs Kater, Geert Kazemier, Johan W. H. Kruimer, Wouter K. G. Leclercq, Christiaan van der Leij, Eric R. Manusama, Mark A. J. Meier, Bram B. van der Meijs, Marleen C. A. M. Melenhorst, Karin Nielsen, Maarten W. Nijkamp, Fons H. Potters, Warner Prevoo, Floris J. Rietema, Alette H. Ruarus, Simeon J. S. Ruiter, Evelien A. C. Schouten, Gian Piero Serafino, Colin Sietses, Rutger-Jan Swijnenburg, Florentine E. F. Timmer, Kathelijn S. Versteeg, Ted Vink, Jan J. J. de Vries, Johannes H. W. de Wilt, Barbara M. Zonderhuis, Hester J. Scheffer, Petrousjka M. P. van den Tol, Martijn R. Meijerink

**Affiliations:** 1Department of Radiology and Nuclear Medicine, Amsterdam UMC, location VUmc, 1081 HV Amsterdam, The Netherlands; s.nieuwenhuizen1@amsterdamumc.nl (S.N.); r.puijk@amsterdamumc.nl (R.S.P.); b.vandenbemd@amsterdamumc.nl (B.v.d.B.); b.geboers@amsterdamumc.nl (B.G.); b.vandermeijs@amsterdamumc.nl (B.B.v.d.M.); m.melenhorst@amsterdamumc.nl (M.C.A.M.M.); a.ruarus@amsterdamumc.nl (A.H.R.); e.schouten@amsterdamumc.nl (E.A.C.S.); f.timmer1@amsterdamumc.nl (F.E.F.T.); j.devries1@amsterdamumc.nl (J.J.J.d.V.); hj.scheffer@amsterdamumc.nl (H.J.S.); 2Department of Surgical Oncology, San Raffaele Hospital, 20132 Milan, Italy; aldrighetti.luca@hsr.it; 3Department of Radiology, Nuclear Medicine and Anatomy, Radboud University Medical Center, 6525 GA Nijmegen, The Netherlands; mark.arntz@radboudumc.nl (M.A.); jurgen.futterer@radboudumc.nl (J.J.F.); janjaap.janssen@radboudumc.nl (J.-J.J.); 4Department of Surgical Oncology, Radboud University Medical Center, 6525 GA Nijmegen, The Netherlands; peter.vandenboezem@radboudumc.nl (P.B.v.d.B.); hans.dewilt@radboudumc.nl (J.H.W.d.W.); 5Department of Radiation Oncology, Amsterdam UMC, location VUmc, 1081 HV Amsterdam, The Netherlands; ame.bruynzeel@amsterdamumc.nl (A.M.E.B.); cja.haasbeek@amsterdamumc.nl (C.J.A.H.); 6Department of Radiology, Leiden University Medical Center, 2333 ZA Leiden, The Netherlands; M.C.Burgmans@lumc.nl; 7Department of Radiology, San Raffaele Hospital, 20132 Milan, Italy; decobelli.francesco@hsr.it; 8Department of Surgical Oncology, Maastricht University Medical Center, 6229 HX Maastricht, The Netherlands; marielle.coolsen@maastrichtuniversity.nl (M.M.E.C.); chc.dejong@mumc.nl (C.H.C.D.); 9Department of Medical Oncology, Amsterdam UMC, location VUmc, Amsterdam, The Netherlands; Oncode Institute, 3521 AL Utrecht, The Netherlands; s.derks@amsterdamumc.nl; 10Department of Radiology, Ziekenhuis Gelderse Vallei, 6716 RP Ede, The Netherlands; DiederikA@zgv.nl (A.D.); grootg@zgv.nl (G.G.); 11Department of Surgical Oncology, Gelre Ziekenhuizen, 7334 DZ Apeldoorn, The Netherlands; p.van.duijvendijk@gelre.nl; 12Department of Surgical Oncology, Medical Center Leeuwarden, 8934 AD Leeuwarden, The Netherlands; hasan.eker@znb.nl (H.H.E.); ermanusama@znb.nl (E.R.M.); 13Department of Surgical Oncology, Amsterdam UMC, location VUmc, 1081 HV Amsterdam, The Netherlands; a.f.engelsman@amsterdamumc.nl (A.F.E.); j.i.erdmann@amsterdamumc.nl (J.I.E.); g.kazemier@amsterdamumc.nl (G.K.); k.nielsen@amsterdamumc.nl (K.N.); r.j.swijnenburg@amsterdamumc.nl (R.-J.S.); Bm.zonderhuis@amsterdamumc.nl (B.M.Z.); mp.vandentol@amsterdamumc.nl (P.M.P.v.d.T.); 14Department of Hepato-Pancreato-Biliary Surgery and Liver Transplantation, University Medical Center Groningen, University of Groningen, 9713 GZ Groningen, The Netherlands; k.p.de.jong@umcg.nl (K.P.d.J.); m.w.nijkamp@umcg.nl (M.W.N.); s.j.s.ruiter@umcg.nl (S.J.S.R.); 15Department of Radiology, University Medical Center Groningen, 9713 GZ Groningen, The Netherlands; g.m.kater@umcg.nl; 16Department of Radiology, Maxima Medical Center, 5504 DB Veldhoven, The Netherlands; Han.Kruimer@mmc.nl; 17Department of Surgical Oncology, Maxima Medical Center, 5504 DB Veldhoven, The Netherlands; W.Leclercq@mmc.nl; 18Department of Radiology, Maastricht University Medical Center, 6229 HX Maastricht, The Netherlands; christiaan.vander.leij@mumc.nl; 19Department of Radiology, Isala Ziekenhuis, 8025 AB Zwolle, The Netherlands; m.a.j.meier@isala.nl (M.A.J.M.); pottersfons@gmail.com (F.H.P.); 20Department of Radiology, OLVG, 1090 HM Amsterdam, The Netherlands; warnerprevoo@me.com; 21Department of Radiology, Noordwest Ziekenhuisgroep, 1815 JD Alkmaar, The Netherlands; f.j.rietema@nwz.nl; 22Department of Radiology, Jeroen Bosch Ziekenhuis, 5223 GZ ‘s-Hertogenbosch, The Netherlands; G.Serafino@jbz.nl; 23Department of Surgical Oncology, Ziekenhuis Gelderse Vallei, 6716 RP Ede, The Netherlands; SietsesC@zgv.nl; 24Department of Medical Oncology, Amsterdam UMC, location VUmc, 1081 HV Amsterdam, The Netherlands; k.versteeg@amsterdamumc.nl; 25Department of Radiology, Medical Center Leeuwarden, 8934 AD Leeuwarden, The Netherlands; ted.vink@znb.nl

**Keywords:** colorectal liver metastases, thermal ablation, microwave ablation, radiofrequency ablation, partial hepatectomy, irreversible electroporation, stereotactic body radiotherapy, resectability criteria, ablatability criteria, consensus guideline

## Abstract

The guidelines for metastatic colorectal cancer crudely state that the best local treatment should be selected from a ‘toolbox’ of techniques according to patient- and treatment-related factors. We created an interdisciplinary, consensus-based algorithm with specific resectability and ablatability criteria for the treatment of colorectal liver metastases (CRLM). To pursue consensus, members of the multidisciplinary COLLISION and COLDFIRE trial expert panel employed the RAND appropriateness method (RAM). Statements regarding patient, disease, tumor and treatment characteristics were categorized as appropriate, equipoise or inappropriate. Patients with ECOG≤2, ASA≤3 and Charlson comorbidity index ≤8 should be considered fit for curative-intent local therapy. When easily resectable and/or ablatable (stage IVa), (neo)adjuvant systemic therapy is not indicated. When requiring major hepatectomy (stage IVb), neo-adjuvant systemic therapy is appropriate for early metachronous disease and to reduce procedural risk. To downstage patients (stage IVc), downsizing induction systemic therapy and/or future remnant augmentation is advised. Disease can only be deemed permanently unsuitable for local therapy if downstaging failed (stage IVd). Liver resection remains the gold standard. Thermal ablation is reserved for unresectable CRLM, deep-seated resectable CRLM and can be considered when patients are in poor health. Irreversible electroporation and stereotactic body radiotherapy can be considered for unresectable perihilar and perivascular CRLM 0-5cm. This consensus document provides per-patient and per-tumor resectability and ablatability criteria for the treatment of CRLM. These criteria are intended to aid tumor board discussions, improve consistency when designing prospective trials and advance intersociety communications. Areas where consensus is lacking warrant future comparative studies.

## 1. Introduction

Colorectal cancer (CRC) is the third most prevalent cancer in the world and, with nearly 881,000 deaths in 2018, the second leading cause of cancer related death [1]. The liver is the most common site of metastases, present at the time of diagnosis in roughly 20% and developed during the course of disease in an additional 40% [2,3,4,5]. Around 40% of patients with colorectal liver metastases (CRLM) have metastatic disease confined to the liver at first discovery. Curative-intent local treatments are currently considered the only realistic treatment options that can provide long-term disease control and cure in a select group of patients [6,7]. Advances in systemic regimens greatly contributed by downstaging patients for liver surgery and/or tumor ablation [8]. Furthermore, it opens a window to identify biologically aggressive fast disseminating cancers that cannot be controlled by local invasive treatments.

Although the eligibility for hepatic resection continues to expand, in approximately 80% upfront surgical excision of all CRLM is not possible [2]. Nowadays, the decision to opt for resection is not only predicated upon tumor-related factors such as size, number, location and distribution, but also upon retaining a sufficient future liver remnant (FLR) [2,9]. Induction systemic therapy for disease that can potentially be downstaged, combined resection plus ablation, portal vein embolization with or without venous deprivation, lobar trans-arterial Yttrium-90 radio-embolization and a variety of two-stage procedures for bilobar disease have greatly contributed to this development [10]. Radiofrequency ablation (RFA) and microwave ablation (MWA) are heat-based thermal ablation modalities, currently adopted as standard of care to treat unresectable small (0-3cm) CRLM [11]. Two recently published systematic reviews and meta-analyses comparing thermal ablation to chemotherapy alone and to partial hepatectomy, both labelled thermal ablation superior to chemotherapy alone but inferior to surgery with regards to overall survival [11,12]. Global guidelines state that thermal ablation should be reserved for unresectable disease. However, in the absence of generally accepted recommendations, the option of thermal ablation as a safe and fair alternative for small deep-seated resectable CRLM has further blurred the definition of resectable disease. Although most superficial, shallow- and deep-seated, small-size CRLM seem to be suitable for thermal ablation, peritumoral vicinity of the common, left or right hepatic bile duct are considered absolute contra-indications as this is associated with an unacceptable risk of inducing biliary tract injuries [13]. Irreversible electroporation (IRE), a predominantly non-thermal ablation technique assumed to spare blood vessels, bile ducts and adjacent organs, engenders ultra-short high-voltage currents that create lethal nanopores in the cell membrane of tumor tissue [14,15,16]. With stereotactic body radiotherapy (SBRT), high radiation doses are delivered to a target volume within the liver, while minimizing collateral damage to healthy surrounding tissue [17,18,19].

Although several clinical staging and classification systems provide prognostic information to predict outcome based on available parameters, and notwithstanding several attempts to postulate resectability criteria, clearly defined and combined resectability and ablatability criteria are absent. As a result, local treatment strategies for liver only metastatic CRC patients are exceedingly heterogeneous and the quality depends upon local expertise and the existence of regional or national referring networks. In light of the increasingly complex patient and disease characteristics and the ever-expanding toolbox of treatment options, there is a necessity to establish criteria that reflect both the technically feasible, the safest and the most effective local treatment option for CRLM patients.

The purpose of this project was to create multidisciplinary resectability and ablatability consensus criteria amongst a large group of experts and to postulate a therapeutic decision model for patients with CRLM based on the highest available evidence levels and classified according to patient, disease and tumor characteristics. 

## 2. Methods

### 2.1. Expert Panel 

Members of the expert panels collaborating in the COLLISION trial [20] (registered at ClinicalTrials.gov NCT03088150), an international phase III randomized controlled trial comparing partial hepatectomy with thermal ablation for small-size resectable CRLM, the COLLISION-XL trial (registered at ClinicalTrials.gov NCT04081168) [21], a multicenter phase II/III randomized controlled trial comparing MWA with SBRT for intermediate-size unresectable CRLM and the COLDFIRE-2 trial [22] (registered at ClinicalTrials.gov *NCT02082782*), a two-center phase IIb prospective clinical trial, first composed a list of patient, disease, tumor and previous treatment characteristics that can potentially influence the decision of the preferred local treatment strategy. Panelists had to fulfill the following requirements: minimum experience of 3 years performing and/or supervising procedures in CRLM patients as surgeons, interventional oncologists or radiation oncologists, having performed and/or supervised over 100 procedures, good clinical practice (GCP) certified and local investigator for at least one of the abovementioned studies. The panel eventually consisted of 19 liver surgeons, 21 interventional radiologists, two radiation oncologists, one technical physician trained to perform ablations and two medical oncologists specialized in colorectal cancer.

### 2.2. Patient, Disease, Tumor and Previous Treatment Characteristics

Potentially decision-affecting patient characteristics assessed were age, Eastern Cooperative Oncology Group (ECOG) performance status [23], American Society of Anaesthesiologists (ASA) physical status classification system [24], underlying liver disease (none, mild or severe) [25] and the Charlson Comorbidity Index (CCI) [25]. The *Disease characteristics* evaluated were synchronous versus metachronous disease and for the latter the time elapsed between primary tumor diagnosis and the first detection of CRLM (taking into account any (neo)adjuvant systemic treatments following bowel surgery), (y)pT-stage and (y)pN-stage for previously resected primaries, (y)cT-stage and (y)cN-stage for potentially resectable and for upfront unresectable primaries, CEA levels, *K-RAS, N-RAS* or *BRAF* mutational status, microsatellite (in)stability, the clustered consensus molecular subtype [26] (CMS1 (microsatellite instability immune), hypermutated, microsatellite unstable and strong immune activation; CMS2 (canonical), epithelial, marked WNT and MYC signaling activation; CMS3 (metabolic), epithelial and evident metabolic dysregulation; and CMS4 (mesenchymal)), the location of the primary tumor (right versus left sided and colon versus rectum), the clustered and validated clinical risk score (CRS) by Fong and colleagues [27] and the modified CRS [28], more recently suggested by the MD Anderson medical center. We classified CRLM into four stages: (1)Stage IVa disease: easily resectable/ablatable requiring minor hepatectomy and/or ablations;(2)Stage IVb disease: difficultly resectable/ablatable requiring major hepatectomy (+/– ablations);(3)Stage IVc disease: initially unresectable/unablatable, but potentially downstageable CRLM where induction systemic therapy and/or future remnant augmentation are appropriate;(4)Stage IVd disease: permanently unresectable/unablatable CRLM in patients ineligible to receive systemic therapy or after unsuccessful downstaging.

Tumor characteristics such as number, size, location (segment, exophytic versus superficial versus deep seated) anatomical relationship to critical structures such as hepatic arteries, portal and systemic veins and the central bile ducts and tumor distribution (scattered or clustered, mono- or bilobar) of CRLM and volume of the FLR were analyzed, as was the preference to opt for less-invasive parenchyma-sparing versus en-bloc major hepatectomy. Further features that potentially impact the therapeutic decision concerned the surgical and medical history, such as abdominal adhesions, recovery from previous abdominal surgery, the objectified response to previous lines of systemic therapy and the number of earlier cycles of chemotherapy and/or biological agent(s). Although trans-arterial therapy such as Yttrium-90 (Y^90^) or Holmium-66 (H^66^) selective internal radiotherapy and trans-arterial chemo-embolization have demonstrated the ability to downstage patients for curative-intent surgery and/or ablation, this sequence as well as radiation segmentectomy, cryo-ablation, laser-induced thermal therapy, needle-based brachytherapy and high-intensity-focused ultrasound for CRLM were considered evolving treatments under investigation and were hence disregarded. Extrahepatic disease was considered off-scope for the current project.

### 2.3. Study Design

We employed the RAND Corporation/University of California Los Angeles Appropriateness Method (RAM) to measure the appropriateness of different local treatment strategies for specific patient, disease and tumor characteristics [29]. In this “modified Delphi” process, experts from multiple disciplines use the available scientific evidence and supplement this evidence with their expert opinions. Each item could be rated on a scale from 1 to 9 (Likert scale), where 1 indicates that the treatment is highly inappropriate (expected harms greatly outweigh the expected benefits) and 9 indicates that it is very appropriate (expected benefits greatly outweigh the expected harms). Within each item the *average* patient was considered. A median score ranging 1-3 means a treatment is inappropriate, 4–6 means it is uncertain (equipoise) and 7–9 means the treatment is considered appropriate. When at least 80% of panelist scored in the same range, the consensus was defined as strong. When 70–80% scored in the same range, the consensus was defined as moderate. Below 70%, a consensus was not reached.

The coordinating committee performed a PubMed literature search on a point-by-point basis in February 2020 for studies in English concerning the treatment of CRLM. Although no formal systematic review was conducted, the search was conformal to the PRISMA guidelines with regards to the information sources used, the search performed and the studies selected (identification, screening and verification by two independent authors). The search can be found in Appendix A. We selected the most relevant papers in order of priority: meta-analysis, systematic review, randomized controlled trials, non-randomized controlled trials, prospective cohort studies, case-control studies, case series, and expert opinions. The levels of evidence of the retrieved articles were independently assessed by the two senior authors (MPvdT, MRM) conformal to the Grading of Recommendations, Assessment, Development and Evaluations (GRADE) system. Discrepancies were resolved by consensus. All relevant characteristics were identified and included in a comprehensive list of (contra-)indications (see Appendix A). The evaluating committee participated in two rating rounds. Before starting the first round, panelists received a list with definitions (see Appendix A) and three articles that they were asked to read as preparation [11,30,31]. In the first round, panelists privately rated the statements using an online questionnaire. The statements that did not achieve strong consensus in round 1 were discussed during a video conference. Panelists could voice their doubts about statements and if deemed necessary by the majority of the panel statement were rephrased. Afterwards the panelists that responded to the first round of the survey received the second and final round of the survey with the statements that did not reach consensus in the first round and the results of the first round were reported back beneath the statements. 

## 3. Results

For rounds 1 and 2, response rates were 44/48 (92%) and 33/44 (75%), respectively. In the first round, consensus was reached for 12/25 statements. Based on discussions during the video conference, 5/13 statements were textually rephrased, and 1 statement was completely rephrased (Appendix A, statement 21). In round 2, strong consensus was reached for 8/13 remaining statements, moderate consensus in 4/13 statements and no consensus in 1/13 statements (see online Appendix A for all statements plus results). Substantiated by the established criteria from the expert panel’s assessments per-patient (Figure 1) and per-tumor (see Figure 2) flowcharts for the treatment of CRLM were created.

### 3.1. Agreements

#### Patient Characteristics

Local therapy should not be withheld from patients based on age alone (evidence level low to moderate—strong consensus) [32,33,34,35]. Partial hepatectomy, thermal ablation, IRE and SBRT are appropriate treatment options for liver only metastatic CRC patients with ECOG ≤2, ASA ≤3 and CCI ≤8; SBRT can be considered for select patients with ECOG 3 (if life expectancy >1 year), ASA 4 or CCI- 9-10 (evidence level low—strong consensus) [36,37,38,39,40,41,42,43,44,45,46,47,48,49,50]. Local therapy is appropriate for patients with no or mild underlying liver disease; for patients with severe underlying liver disease the risks of the procedure do not outweigh the benefits (evidence level low—moderate consensus) [51,52].

### 3.2. Disease Characteristics

For patients with stage IVa and stage IVb disease, defined as disease requiring minor hepatectomy/ablations versus disease requiring major hepatectomy, the appropriate treatment is upfront surgery and/or ablation without peri-procedural systemic therapy (evidence level high— strong consensus) [53,54]. However in stage IVb disease, there are two exceptions where pre-procedural systemic therapy is indicated: (1) when downsizing of CRLM is likely to reduce the procedural risk (induction systemic therapy) and (2) in chemo-naïve patients that did not have CRLM at time of diagnosis of the primary tumor, who developed multiple CRLM that would require major hepatectomy within six months, indicating aggressive and fast disseminating tumor biology (neo-adjuvant systemic therapy). Neo-adjuvant systemic therapy would prevent patients from receiving futile invasive local therapy if early disease progression under systemic therapy is present (evidence level low—strong consensus) [55,56,57,58]. In stage IVc, defined as initially unresectable but potentially downstageable CRLM, induction systemic therapy is appropriate until (a) curative-intent local treatment has become possible or (b) when additional downsizing will not (further) decrease procedural risk (evidence level high—strong consensus) [6,7,8,59,60,61,62]. Stage IVd disease, defined as permanently unsuitable for curative intent local therapy, should be reserved for unresectable/unablatable patients who fail downstaging chemotherapy and for unresectable/unablatable patients who do not qualify for downstaging systemic therapy (evidence level moderate—strong consensus) [63]. Orthotopic liver transplantation (OLT) can be considered for highly select patients permanently unsuitable for local treatment, at the prerequisite that a sufficient number of liver allografts is available and merely in the setting of prospective registries and/or trials. Although promising prognosticators, liquid and tissue biomarkers as well as validated classification systems such as the CRS by Fong and colleagues and the modified CRS have not yet shown additive value in the selection of specific local treatment options (no evidence—strong consensus) [27,28,64,65,66]. Patients cannot be disqualified for local therapy based on a certain number and/or size of CRLM; the upper limit is defined by respecting the thresholds of the estimated future liver remnant volume and/or function (evidence level low—consensus strong) [67,68,69,70]. 

### 3.3. Tumor Characteristics

Partial hepatectomy is the standard of care for liver only colorectal metastases (evidence level low—consensus strong) [71,72,73,74,75]. However, in patients with a poor general health status (ECOG 2 *and* ASA 3 or CCI 5-8) thermal ablation can be considered as an alternative treatment option (evidence level low—consensus high) [9,11,76,77,78]. For small-size and resectable CRLM that are deep seated requiring major hepatectomy, thermal ablation is the appropriate treatment (evidence level low—consensus strong) [79]. Unresectable CRLM ≤3cm should be treated by thermal ablation. Thermal ablation can be considered for unresectable CRLM 3–5 cm when further downsizing systemic therapy is unfeasible (evidence level moderate to high—consensus strong) [9,11,12,76,80,81,82]. IRE and SBRT can be considered for patients with unresectable and not thermally ablatable CRLM (evidence level low—consensus moderate) [14,15,16,83,84,85]. IRE is appropriate for perihilar and/or perivascular CRLM ≤3 cm, and 3–5 cm if further downsizing therapy is unfeasible (level of evidence low—consensus moderate) [14,15,16,84]. SBRT can be considered for select patients with a limited disease burden (≤3 CRLM) and tumors ≤5 cm, at the prerequisite that an ablative dose can be delivered without jeopardizing liver function or other organs at risk and that ECOG is ≤3, ASA is ≤4 or CCI is ≤10 (level of evidence low—consensus strong) [83].

Three distinct types of CRLM that are eligible for local treatment emerged: 1) CRLM that should be resected (type I), 2) CRLM that should be treated with thermal ablation (type II) and 3) CRLM eligible for non-thermal ablation (type III). The following subtypes were categorised: CRLM that are unablatable but suitable for resection (type Ia), CRLM that are resectable and ablatable with a preference for resection (type Ib), CRLM that are resectable and ablatable with a preference for thermal ablation (type IIa) and CRLM that are considered unsuitable for resection (type IIb). The last category entails the anatomically unresectable and not thermally ablatable CRLM, eligible for IRE or SBRT (type IIIa) and the patients with a very poor general health status but fair life expectancy >1 year, eligible for SBRT (type IIIb) (see Table 1).

### 3.4. Intertumor or Clustered Dependency Characteristics 

When multiple (≥3) deep-seated CRLM (+/− other CRLM) are present in a single lobe, with or without limited contralateral disease, and remnant liver volume and/or function are adequate, single-session partial hepatectomy is the appropriate treatment (evidence level low—consensus moderate). When multiple (≥3) deep-seated and small-size CRLM (+/− other CRLM) are present in both lobes, and remnant liver volume and/or function would be inadequate, both a “2-stage hepatectomy” and a 1-stage “chip-and-burn” procedure (thermal ablation of the deep-seated small-size CRLM and resections of the other CRLM) can be considered (no consensus on preferred method) [86,87,88].

### 3.5. Treatment Characteristics 

The anatomical contra-indications for partial hepatectomy are as follows: (1) inability to obtain R0 margins (R1 margins in case of vascular involvement), (2) inability to preserve a sufficient FLR volume and/or function, (3) inability to preserve the dual blood supply and the venous and biliary drainage from the FLR and (4) inaccessibility of the abdominal cavity due to excessive abdominal adhesions (strong consensus) [30,68,89]. The anatomical contra-indications for thermal ablation are as follows: (1) peri-tumoral vicinity (<10 mm) of the common, left or right hepatic bile duct or (2) peri-hepatic critical structures that cannot be distanced using surgical or interventional dissection methods, (3) the abutment or encasement of a single remaining major portal or systemic vein following surgery and (4) invasion of the free wall of the inferior caval vein. The maximum tumor size is 3 cm, although thermal ablation *can* be considered for 3–5 cm unresectable CRLM after failure to (further) downsizing with systemic therapy (strong consensus) [90]. The contra-indications for IRE are CRLM >5 cm, ventricular arrhythmias, cardiac stimulation devices and congestive heart failure (strong consensus) [22]. Contra-indications for SBRT are >3 CRLM and the inability to deliver an ablative radiation dose without jeopardizing liver function and adjacent organs or structures at risk (moderate consensus) [91].

## 4. Discussion

This article describes the development of a multidisciplinary expert panel consensus-based treatment algorithm for patients with liver-only colorectal cancer metastases. Given the rapidly changing landscape and the multitude of novel local, regional and systemic treatments, a guideline with directive rules of decision cannot be expected to be truly complete or to remain permanent. Similarly, the items of consensus do not encompass the full spectrum of interpatient variability and individual exceptions. The postulated agreements are contemporary and intended to guide multidisciplinary team meetings, optimize future prospective studies and improve intersociety communications.

Although several attempts to propose resectability criteria have been reported, combined resectability and ablatability criteria have not been postulated [68,89,92]. An early effort to classify CRLM patients was proposed by Nordlinger and colleagues in ‘the European Colorectal Metastases Treatment Group staging system’, and subdivided patients into resectable (M1a), potentially resectable (M1b) and metastases unlikely to become resectable (M1c) [89]. More recently the ESMO consensus guidelines stated that “the best local treatment should be selected from a ‘toolbox’ of procedures according to disease localization, treatment goal, treatment-related morbidity and patient-related factors such as comorbidities and age” [30]. However, the ESMO guidelines do not define the appropriate local treatment option based on specific patient, disease and tumor characteristics.

Nineteen surgeons, 21 interventional radiologists, two radiation oncologists, two medical oncologists and one technical physician reached strong consensus on 20/25 statements and moderate consensus on another four statements. With respect to patient characteristics, strong consensus was reached for the ECOG performance status, ASA score and CCI thresholds; moderate consensus was reached on the statement that local treatment should be withheld from patients with severe underlying liver disease. Regarding disease characteristics, strong consensus was eventually reached on all items (subgroup definitions, discouraging the use of prognostic biomarkers, and size and number of CRLM as predictive parameters). Considering tumor characteristics, strong to moderate consensus was reached on the majority of items regarding the preferred local treatment per specific anatomical location. Although strong consensus was reached regarding hepatectomy as the preferred treatment option for patients with multiple deep-seated CRLM in one lobe, no consensus was reached for patients with multiple deep-seated CRLM in both lobes. After some textual adjustments following round 1, a strong consensus was eventually reached on the contra-indications for partial hepatectomy, thermal ablation and IRE and moderate consensus on SBRT. The lower consensus regarding SBRT to treat CRLM may be a result of the paucity of disease-specific and/or comparative studies with hard oncological endpoints throughout the literature [93,94,95]. However, a fair amount of studies did show promising results regarding toxicity and local control to treat tumors within the liver and the results seem to be improving [18,19,96]. Given the partially overlapping indications, the exact role of IRE and SBRT in the treatment of unresectable and not thermally ablatable CRLM needs to be clarified in future prospective studies. 

An equal distribution amongst panelists existed for a ‘2-stage hepatectomy’ versus a ‘single-session chip-and-burn procedure’ in this subgroup of patients with advanced disease. Although the historical gold standard represents a 2-stage hepatectomy with future liver remnant augmentation, a trend towards parenchyma sparing procedures exists [86]. As the complication rate and outcomes of these complex procedures strongly depend on operator experience, in the absence of comparative studies it is recommended to leave the decision up to local expertise. 

In the prospective SECA-I and -II trials, 36 patients with unresectable colorectal cancer liver only metastases underwent liver transplantation. Under very strict selection criteria, promising results were reported with 5-year overall survival from transplantation reaching 83% [97,98,99,100]. These results led to conditional and preliminary acceptation of liver transplantation in a highly select subgroup of patients with CRLM permanently unsuitable for local treatment in several countries. However, given the liver allograft shortages in most regions, it is recommended to strictly adhere to the national or regional selection criteria and to merely offer this treatment in the setting of prospective registries or trials. 

Hopefully, in the near future the outcomes of studies such as the phase III non-inferiority randomized controlled COLLISION trial that seeks to compare partial hepatectomy to thermal ablation for resectable CRLM, the COLLISION-XL trial that compares MWA to SBRT for intermediate-size unresectable CRLM, an RCT in Denmark (*NCT03654131*) and an RCT in Italy (*NCT02820194*) for CRLM <4 cm that compare SBRT to MWA, and the COLDFIRE-2 trial that investigates the role of IRE for perihilar and perivascular CRLM will be able to shed light on these issues [20,21,22].

The high number of expert participants, the multidisciplinary approach, the good response rates and the high level of strong consensus add value to this consensus guideline. Although the items of agreement aimed to cover both patients with and without a history of local therapy for CRLM, the reported evidence mainly focused on the first metastatic episode. Hence, the outcomes are less applicable to patients with distant recurrences in the liver, given the presumed higher complexity of surgical procedures in a previously accessed abdominal cavity and given the often reduced liver remnant volume [101]. One potential shortcoming of our study was the underrepresentation of some professions in the multidisciplinary expert panel (two radiation oncologists, two medical oncologists and one technical physician, compared to 19 surgeons and 21 interventional radiologists), which potentially weakens the generalizability of the agreements. Furthermore, the fact that the work represents opinions from predominantly Dutch physicians may have contributed to a skewed view on the role of peri-procedural systemic therapy for CRLM as this is still controversial. Another drawback is that despite our efforts to postulate clear criteria based on the anatomical location of CRLM, the assessment of the specific location (e.g., exophytic, superficial, shallow and deep seated) remains largely subjective. 

In this consensus guideline we did not discuss the approach of surgery or ablation (e.g., open, laparoscopic, robot assisted and percutaneous). We feel this choice should be made by the specialists performing the procedure, as this decision depends on local knowledge, skills and resources.

Several treatments for the local or regional treatment of CRLM were considered ‘off-scope’, such as cryo-ablation, Y^90^ or H^66^ selective internal radiation therapy, trans-arterial chemo-embolization, high-intensity-focused ultrasound, laser-induced thermotherapy, hepatic artery infusion (pump) chemotherapy and brachytherapy. Although all represent promising techniques, research is ongoing and the exact role in the treatment of CRLM will need to be defined in future prospective controlled studies.

## 5. Conclusions 

This paper provides a framework of key opinion leader agreements concerning resectability and ablatability criteria for CRLM. We encourage all of our colleagues to adopt the recommendations outlined in order to aid tumor board discussions, to improve consistency in the design of future prospective trials and to advance intersociety communications. 

## Figures and Tables

**Figure 1 cancers-12-01779-f001:**
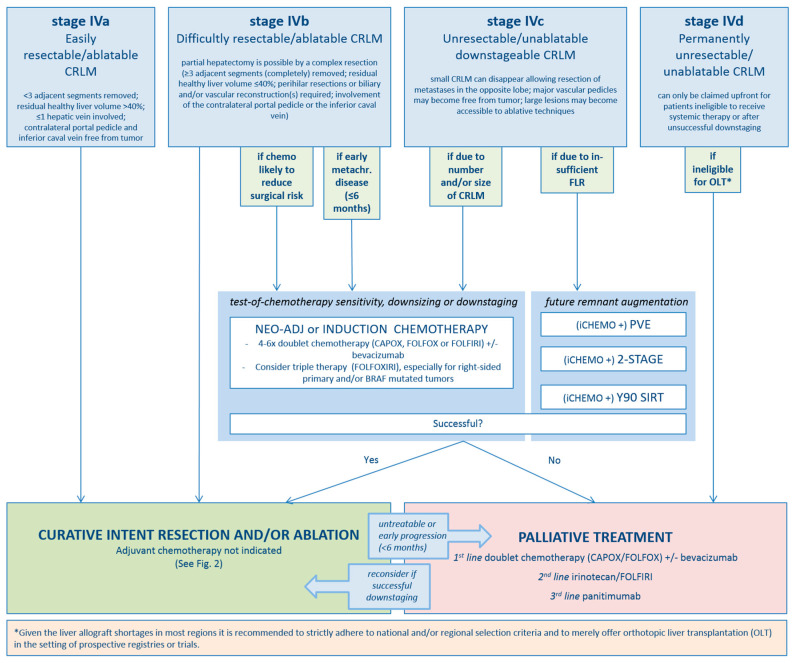
Per-patient flowchart.

**Figure 2 cancers-12-01779-f002:**
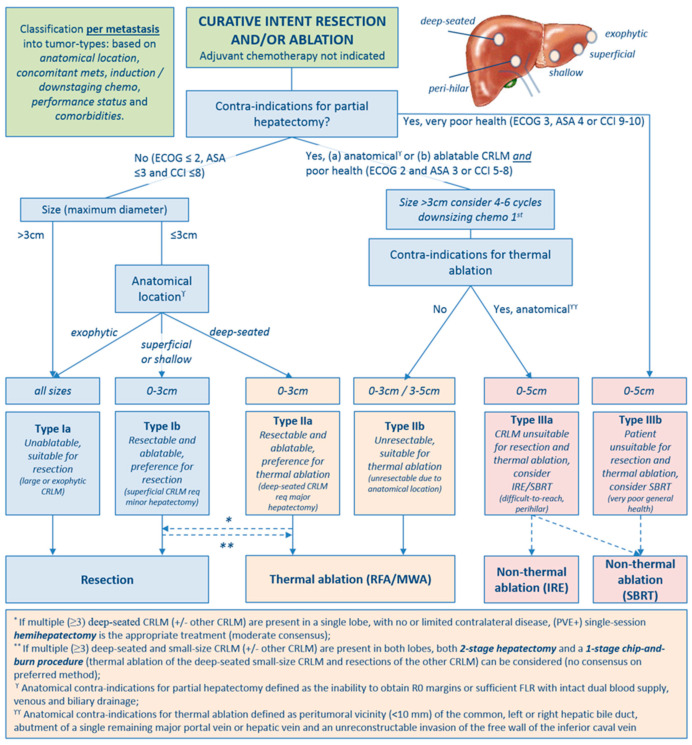
Per-tumor flowchart.

**Table 1 cancers-12-01779-t001:** Type of CRLM for which locoregional therapy should be considered.

Type I *Resection*	Type II *Thermal ablation RFA/MWA*	Type III *Non-thermal ablation IRE/SBRT*
**Ia: Unablatable, suitable for resection**	**IIa:** Resectable and ablatable, preference for thermal ablation	**IIIa:** Unresectable and unablatable, consider IRE or SBRT
**Ib: Resectable and ablatable, preference for resection**	**IIb:** Unresectable, suitable for thermal ablation	**IIIb:** Unresectable and unablatable, consider SBRT

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
