# Peer review of "Resectability and Ablatability Criteria for the Treatment of Liver Only Colorectal Metastases: Multidisciplinary Consensus Document from the COLLISION Trial Group"

_cancers, 2020, doi:10.3390/cancers12071779_

Round 1

Reviewer 1 Report

Dear editors,

I have read with interest the paper “Resectability and ablatability criteria for the treatment of liver only colorectal metastases: multidisciplinary consensus document from the COLLISION Trial Group”. The large panel of experts is to be commended for producing such a wide-ranging consensus about the treatment of liver-only colorectal metastases.

The questions addressed by the research are of great interest and the quality of writing confirms that the authors put a huge effort in producing a basis from which to advance in this field of clinical research. Accordingly I consider the paper worth publishing.

Nonetheless, I have both minor and major concerns to which I would like to draw the authors’ attention.

Minor concerns:

  1. At page two, section “keywords”: there is a typo in writing “partial hepatectomy”.
  2. I suggest against the use of the plural “therapies”. The singular “therapy” or the plural “treatments” are preferable.
  3. I assume that the technical physician is a technical physicist. Am I right?

Major concerns:

  1. The literature search strategy and criteria are lacking. Eligibility criteria are not reported and the research is not documented, for example in the form of a PRISMA flowchart. At page 5, the sentence “The coordinating committee performed a PubMed literature search […]” is insufficient for a work that aims to produce recommendations.
  2. The methodology for grading the retrieved evidence is absent. The level of evidence is stated in the online appendix A, but the whole process of assessing and reporting in tables the certainty of evidence, following, for example, the GRADE system, is missing.
  3. I disagree with the premise that liver transplantation for nonresectable liver-only colorectal metastases can be disregarded as a merely experimental treatment, after the extensive experience reported in the SECA studies. Accordingly, the panellists should integrate this option in their algorithm – especially for patients who are fit but harbour unresectable lesions – in order to increase the comprehensiveness of their consensus.

The re-elaboration of the whole paper in the form of a systematic review and meta-analysis with the aim of producing solid guidelines might transcend the purpose of this work. I leave that decision up to the editors. Should the paper be accepted in this form, I feel that it would be more appropriate to carefully remove any reference to the concepts of “recommendation” from the manuscript and leave just the concept of “consensus” or “agreement” on definitions.

Reviewer 2 Report

The authors propose guidelines for the local control of metastases in patients having liver-only metastases from colorectal cancer. They used a RAND-appropriateness-method to reach a consensus among expert from specialties involved in the management of colorectal liver metastases.

The paper is well-written and provides a useful tool in the decision-making process. The question of peri-operative chemotherapy in resectable CRLM is still debatable though.

I have some comments:

1 – Could the authors add the maximum size of the CRLM targeted by SBRT? In the abstract, it is specified that both IRE and SBRT can be used for CRLM of 0-5cm (when unresectable) but not in the text

P8, 3.3

IRE is appropriate for perihilar and/or perivascular CRLM ≤3cm, and 3-5 cm if further downsizing therapy is unfeasible (level of evidence low – consensus moderate)[14-16,84]. SBRT can be considered for select patients with a limited disease burden (≤3 CRLM), at the prerequisite that an ablative dose can be delivered without jeopardizing liver function or other organs at risk and that ECOG is ≤3, ASA is ≤4 or CCI is ≤10 (level of evidence low – consensus strong)[83].

2 – The authors considered that prognostic biomarkers cannot preclude patients for local treatment. Meanwhile, they considered early metachronous as an indicator of aggressive tumor biology.

  1. How do the authors consider synchronous disease? One could argue that it indicates aggressive tumor biology
  2. For several medical oncologist, in case of early oligo metachronous CRLM, up-front surgery or ablation are proposed in patient who received oxaliplatin-based adjuvant chemotherapy for the primary tumor, arguing that CRLM appearing while on adjuvant chemotherapy are chemo-resistant

3 – The main drawback of this study is the underrepresentation of medical oncologist (only one). This limit has been fairly discussed by the authors.

Round 2

Reviewer 1 Report

I thank the authors for their appropriate explanations and corrections. The article can be published in its current form.